# Input-dependent regulation of excitability controls dendritic maturation in somatosensory thalamocortical neurons

Laura Frangeul [1], Vassilis Kehayas [1], Jose V. Sanchez-Mut [2], Sabine Fièvre[1], K. Krishna-K[1,3], Gabrielle Pouchelon[1,4], Ludovic Telley [1], Camilla Bellone[1], Anthony Holtmaat[1], Johannes Gräff[2], Jeffrey D. Macklis [5] & Denis Jabaudon[1,6]

Input from the sensory organs is required to pattern neurons into topographical maps during development. Dendritic complexity critically determines this patterning process; yet, how signals from the periphery act to control dendritic maturation is unclear. Here, using genetic and surgical manipulations of sensory input in mouse somatosensory thalamocortical neurons, we show that membrane excitability is a critical component of dendritic development. Using a combination of genetic approaches, we find that ablation of $N$-methyl-D-aspartate (NMDA) receptors during postnatal development leads to epigenetic repression of Kv1.1-type potassium channels, increased excitability, and impaired dendritic maturation. Lesions to whisker input pathways had similar effects. Overexpression of Kv1.1 was sufficient to enable dendritic maturation in the absence of sensory input. Thus, Kv1.1 acts to tune neuronal excitability and maintain it within a physiological range, allowing dendritic maturation to proceed. Together, these results reveal an input-dependent control over neuronal excitability and dendritic complexity in the development and plasticity of sensory pathways.

[1] Department of Basic Neurosciences, University of Geneva, 1211 Geneva 4, Switzerland. [2] Brain Mind Institute, School of Life Science, Ecole Polytechnique Fédérale de Lausanne, 1015 Lausanne, Switzerland. [3] Department of Physiology, Yong Loo Lin School of Medicine, National University of Singapore, Singapore 117593, Singapore. [4] Department of Neurobiology, Harvard Medical School, Boston, MA 02115, USA. [5] Department of Stem cell and Regenerative Biology, Center for Brain Science, Harvard University, Cambridge, MA 02138, USA. [6] Clinic of Neurology, Geneva University Hospital, 1211 Geneva 14, Switzerland. Correspondence and requests for materials should be addressed to D.J. (email: denis.jabaudon@unige.ch)

Neuronal morphology is critical to allow topographical mapping of the sensory periphery along input pathways. Regulation of dendritic complexity is particularly important in this process, as the expanse of the dendritic tree determines which subset of inputs a given neuron can respond to. During development, dendritic maturation and associated topographical mapping require appropriate input from the periphery[1–3]. For example, in the mouse somatosensory system, functional ablation of N-methyl-D-aspartateNMDA receptors (NMDARs), which are normally activated by sensory input, leads to disrupted barrel map formation[4–8]. Similarly, severing the infraorbital nerve, which carries input from the whiskers, disrupts neuronal patterning at all relay stations of the whisker-to-cortex pathway[9, 10]. Although both procedures disturb sensory mapping, the nature and specificity of the molecular/cellular processes at play in each of these conditions are poorly understood.

Somatosensory thalamocortical neurons of the ventroposterior medial nucleus (VPM) of the thalamus are well suited to study this question: they constitute a homogenous population of somatotopically organized neurons, which respond to whisker inputs and project to the primary somatosensory cortex[11]. During development, NMDARs are important for synaptic transmission at VPM neurons[12] and mosaic deletion of the NMDAR essential subunit Grin1 impairs synaptic maturation[8]. Likewise, functional elimination of trigeminothalamic synapses at VPM neurons is required for somatotopic refinement and is disturbed by sensory deprivation[13].

Using VPM neurons as a model population to study the input-dependent mechanisms underlying dendritic maturation, here we show that developmental genetic inactivation of NMDARs leads to an epigenetic repression of Kv1.1, a potassium channel regulating membrane excitability. As a result, NMDAR-lacking VPM neurons are hyperexcitable and display impaired dendritic maturation. Similarly, surgical disruption of whisker input by section of the infraorbital nerve leads to neuronal hyperexcitability and impaired dendritic maturation. Remarkably, over-expression of Kv1.1 is sufficient to enable dendritic development in both cases, revealing that neuronal excitability is a critical

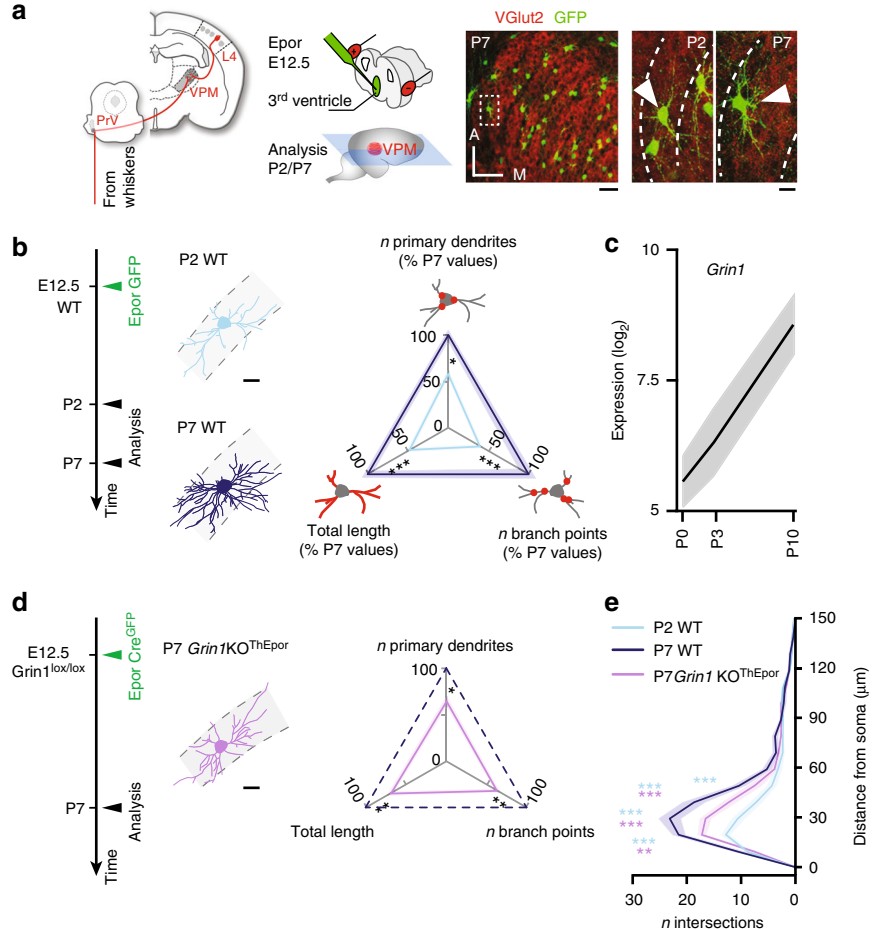

**Fig. 1** NMDAR activation controls the dendritic maturation of VPM neurons during postnatal development. **a** Left: somatosensory thalamocortical neurons of the ventroposterior medial nucleus (VPM) of the thalamus respond to whisker inputs and project to the primary somatosensory cortex. Center: experimental time course and schematic representation of the labeling technique. Right: GFP+ neurons are visible in horizontal sections of the VPM in the low magnification photomicrograph and VGlut2 labeling allows delineation of the barreloids. Scale bar: 100 μm. High magnification of two VPM neurons at P2 and P7 (arrowheads) with white dotted lines delineating barreloids. Scale bar: 20 μm. **b** Dendritic complexity of VPM neurons increases during development (P2 WT $n = 11$ from 2 mice, P7 WT $n = 10$ from 2 mice). Three-axis representation of primary dendrites, number of branch points and total dendritic length. Values are expressed as a percentage of WT P7 VPM neurons values in this and subsequent panels. See Supplementary Fig. 2 for related data. Scale bar: 20 μm. **c** Grin1 expression increases during development. **d** Dendritic maturation is impaired in Grin1 loss-of-function neurons (P7 Grin1KO[ThEpor] $n = 16$ from 3 mice). Scale bar: 20 μm. **e** Sholl analysis of dendritic complexity. PrV, principal trigeminal nucleus; WT, wild-type. One-way ANOVA with Tukey's post-hoc test for all statistical tests relating to dendritic complexity, except for Sholl analyses for which a two-way ANOVA with Tukey post-hoc test was used. *$P < 0.05$, **$P < 0.01$, ***$P < 0.001$; NS, not significant

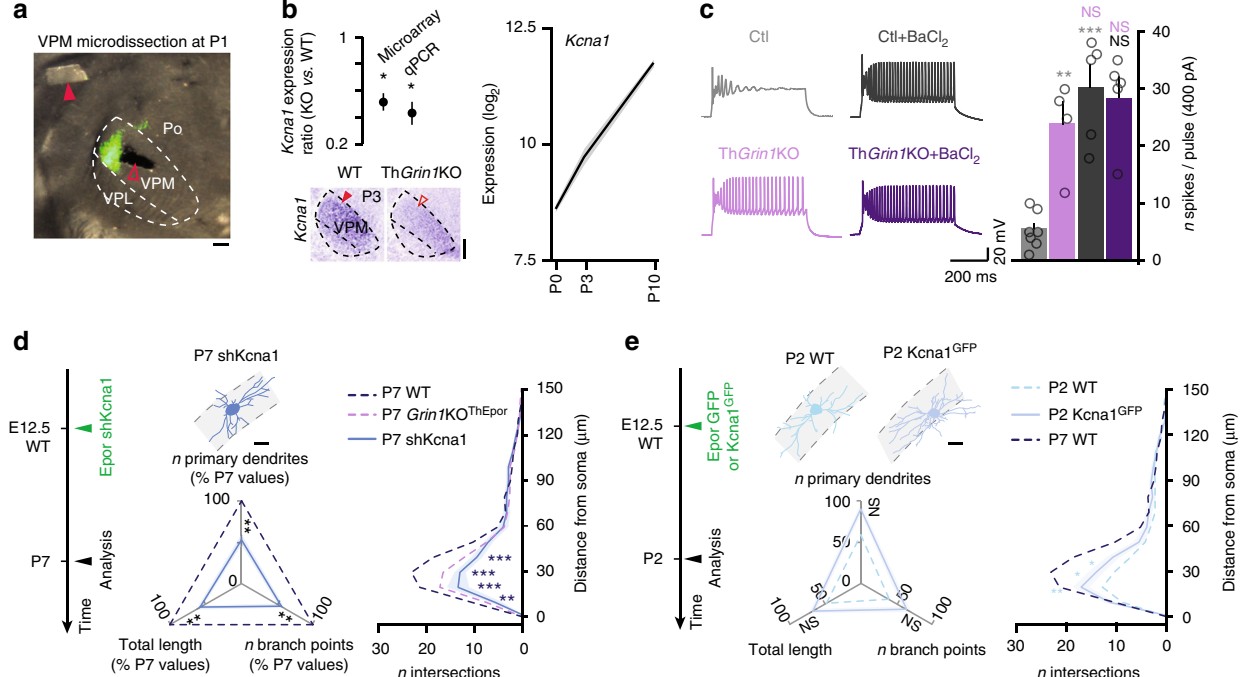

**Fig. 2** Kcna1 is a downstream target of NMDARs and controls neuronal excitability and dendritic maturation. **a** Illustrative microdissection of VPM nucleus at P1. VPM nucleus was identified by retrograde labeling from S1 (green labeling). Red arrowheads show the microdissected specimen and its original location. Scale bar: 100 μm. **b** Top left: Kcna1 expression is decreased in ThGrin1KO ($n = 3$). Red arrowheads show the labeling in VPM. Student's t-test, *$P < 0.05$. Bottom left: in situ hybridization shows decreased Kcna1 expression in ThGrin1KO VPM ($n = 3$). Scale bar: 200 μm. Right: Kcna1 expression increases during development. **c** At P15, ThGrin1KO VPM neurons are hyperexcitable; this occludes the effects of the K$^+$ channel blocker BaCl$_2$. (Ctl $n = 7$, ThGrin1KO $n = 4$, Ctl + BaCl$_2$ $n = 5$, ThGrin1KO + BaCl$_2$ $n = 5$). One-way ANOVA with Tukey's post-hoc test, **$P < 0.01$, ***$P < 0.001$, NS, not significant. **d** At P7, dendritic maturation is impaired by Kcna1 down-regulation (P7 shKcna1 $n = 8$ from 2 mice). P7 WT and P7 Grin1KO$^{ThEpor}$ data reported from Fig. 1b, d. Scale bar: 20 μm. **e** At P2, dendritic maturation is increased by overexpression of Kcna1 (P2 Kcna1$^{GFP}$ $n = 14$ from 4 mice). P2 WT and P7 WT data reported from Fig. 1b. Scale bar: 20 μm. WT, wild-type. One-way ANOVA with Tukey's post-hoc test for all statistical tests relating to dendritic complexity, except for Sholl analyses for which a two-way ANOVA with Tukey's post-hoc test was used. *$P < 0.05$, **$P < 0.01$, ***$P < 0.001$; NS, not significant

determinant of dendritic complexity during assembly of sensory pathways.

## Results

**NMDAR controls the dendritic maturation of VPM neurons.** We characterized the dendritic development of VPM neurons during the first postnatal week in mice using in utero electroporation of a green fluorescent protein (GFP)-expressing plasmid to label individual neurons[14] (Fig. 1a). This approach revealed an increase in dendritic complexity (as assessed by measuring the number of primary dendrites, branch points, and total dendritic length) between P2 and P7, a time at which whisker input becomes functional and pups experience their first extra-uterine stimuli (Fig. 1b, e, and Supplementary Figs. 1a, b and 2).

NMDARs have been implicated in neuronal patterning, and act in part through dendritic changes[15–17]. Suggesting that NMDARs also have a role in dendritic maturation in the VPM, a developmental transcriptional analysis of VPM neurons during this time period[18] showed that expression of the NMDAR essential subunit Grin1 increases in parallel with dendritic complexity (Fig. 1c; $P = 0.016$ for P0 vs. P10, Student's t-test). Given this correlation, we directly examined whether NMDARs control dendritic maturation. For this purpose, we genetically ablated these receptors in VPM neurons using electroporation of a Cre$^{GFP}$ plasmid in Grin1$^{lox/lox}$ mice (referred to as Grin1-KO$^{ThEpor}$), which prevents expression of GRIN1. In the absence of NMDARs, postnatal dendritic maturation of VPM neurons was impaired, as shown by reduced dendritic complexity

by P7 (Fig. 1d, e, and Supplementary Figs. 1c and 2). Of note, in contrast to L4 neurons in cortical barrels[19, 20], the dendritic tree of VPM neurons is not typically confined to a single barreloid[21–23], which remained the case in the absence of NMDARs.

These results indicate that NMDARs control the postnatal dendritic maturation of VPM neurons.

**Kv1.1 controls dendritic maturation.** We next investigated the transcriptional mechanisms controlling dendritic maturation downstream of NMDARs. For this purpose, we generated transgenic mice which lacked Grin1 in VPM neurons (and, to a lesser extent, in the posterior and lateral geniculate nuclei), by crossing Sert$^{Cre}$ mice with Grin1$^{lox/lox}$ mice (henceforth referred to as ThGrin1KO mice; Supplementary Figs. 3a, b). As previously reported, ThGrin1KO mice lacked whisker-specific patterning in the VPM (barreloids). This is similar to what is following neonatal infraorbital nerve section (IONS), which prevents input from the whiskers from reaching the hindbrain[7, 9, 10] (Supplementary Fig. 3c).

We investigated the transcriptional targets of NMDARs by microdissecting VPM neurons in control and ThGrin1KO mice at P1, when dendrites are beginning to extend[11] (Fig. 2a). Among differentially expressed genes, Kcna1 stood out as being both regulated by NMDARs and developmentally regulated[18] (i.e., significantly increased between P0 and P10, Student's t-test; ***$P < 0.001$) (Fig. 2b, Supplementary Fig. 3d, and Supplementary Table 1).

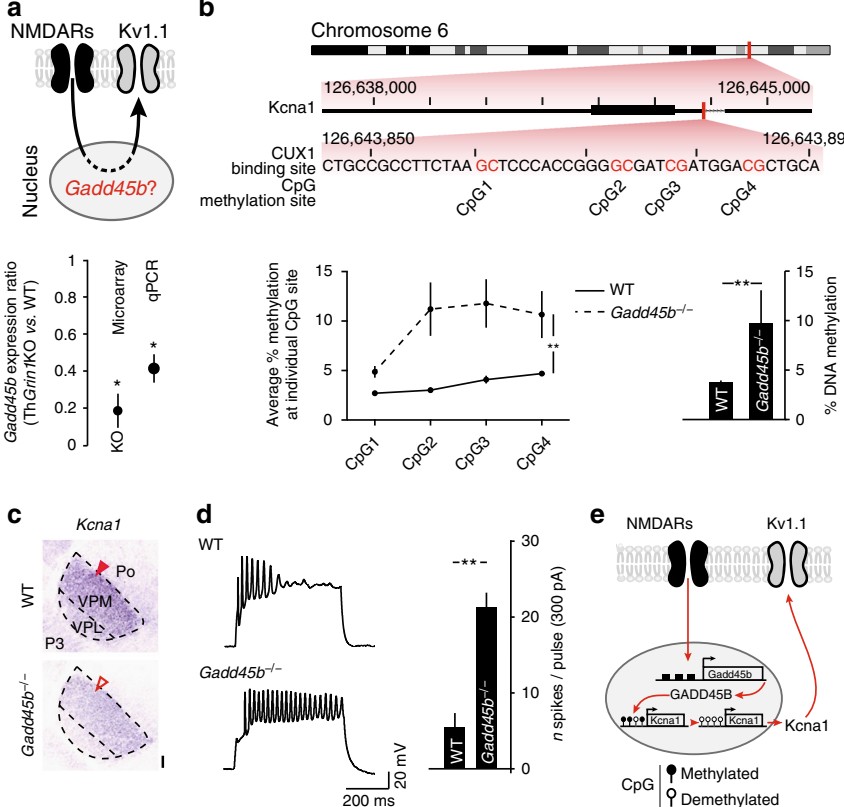

**Fig. 3** GADD45B epigenetically regulates Kv1.1 expression. **a** Top: working hypothesis. Bottom: *Gadd45b* expression is decreased in Th*Grin1*KO VPM neurons. Student's *t*-test, *$P < 0.05$, **$P < 0.01$. **b** DNA methylation of the *Kcna1* promoter region is increased in Gadd45b$^{-/-}$ VPM neurons. Top: schematic representation of the *Kcna1* promoter region. Bottom: bisulfite sequencing results. Kruskal–Wallis test, **$P < 0.01$. **c** In situ hybridization shows decreased *Kcna1* expression in Gadd45b$^{-/-}$ VPM ($n = 3$). Red arrowheads show the labeling in VPM. Scale bar: 100 μm. **d** Gadd45b$^{-/-}$ VPM neurons are hyperexcitable. Student's *t*-test, **$P < 0.01$. **e** Summary of the findings

The gene product of *Kcna1* is Kv1.1, a potassium channel that reduces dendritic excitability[24, 25]. As a consequence, decreased expression of *Kcna1* in Th*Grin1*KO VPM neurons should result in increased neuronal excitability. Confirming this prediction, whole-cell patch-clamp recording of Th*Grin1*KO VPM neurons in acute slices revealed an increase in spike numbers in response to depolarizing current injections, which was replicated by pharmacological blockade of K$^+$ channels in wild-type VPM neurons (Fig. 2c, Supplementary Fig. 4, and Supplementary Table 2). Thus, developmental loss of NMDAR function leads to increased excitability of VPM neurons *via* decreased potassium conductances.

In retinal ganglion cells, Kv1.1 regulates the initial development of dendritic arbors through control of membrane excitability[26]. To examine whether Kv1.1 likewise regulates dendritic complexity in VPM neurons, we downregulated *Kcna1* using in utero electroporation of a small hairpin RNA (shRNA). This led to impaired dendritic maturation with reduced complexity by P7, as we had observed following genetic ablation of NMDARs (Fig. 2d, and Supplementary Figs. 1d, i, 2, and 5). Supporting a role for *Kcna1* expression dynamics in the normal dendritic development of VPM neurons, early overexpression of this gene led to a developmental increase in dendritic complexity (Fig. 2e, and Supplementary Figs. 1e and 2). Together, these results suggest that *Kcna1*, a downstream target of NMDARs, controls neuronal excitability and postnatal dendritic maturation.

**NMDARs epigenetically regulate Kv1.1 via *Gadd45b* expression.** We next investigated the molecular mechanism through which NMDARs regulate Kv1.1 expression. *Gadd45b* is an immediate early gene induced by NMDAR activation. Its gene product GADD45B is responsible for activity-induced DNA demethylation and associated gene de-repression in hippocampal neurons[27]. We identified *Gadd45b* amongst the top genes that were down-regulated following loss of NMDARs in VPM neurons, suggesting that a similar pathway might be active in these neurons (Fig. 3a and Supplementary Table 1). *Gadd45b*$^{-/-}$ mice showed normal whisker patterning (Supplementary Fig. 6 and see Discussion). Using bisulfite pyrosequencing of *Gadd45b*$^{-/-}$ VPM neurons to identify changes in DNA methylation marks, we found an increase in the methylation of *Kcna1* promoter region (Fig. 3b). This epigenetic repression was associated with decreased expression of the *Kcna1* transcript, as demonstrated by in situ hybridization (ISH) (Fig. 3c). Consistent with this decreased expression of Kv1.1, *Gadd45b*$^{-/-}$ neurons were more excitable than their wild-type counterparts when challenged with depolarizing current injections (Fig. 3d). Thus, NMDARs regulate neuronal excitability *via* *Gadd45b*-mediated epigenetic de-repression of Kv1.1 (Fig. 3e).

**VPM neurons are hyperexcitable following IONS.** The findings above suggest a model in which Kv1.1 acts to maintain neuronal activity within a physiological range during postnatal development. When NMDARs are present and active, *Gadd45b* is expressed and the *Kcna1* promoter is demethylated, allowing Kv1.1 expression and decreased neuronal excitability. In the absence of NMDAR activation, the *Kcna1* promoter remains methylated and Kv1.1 expression is repressed, allowing a compensatory increase in neuronal excitability.

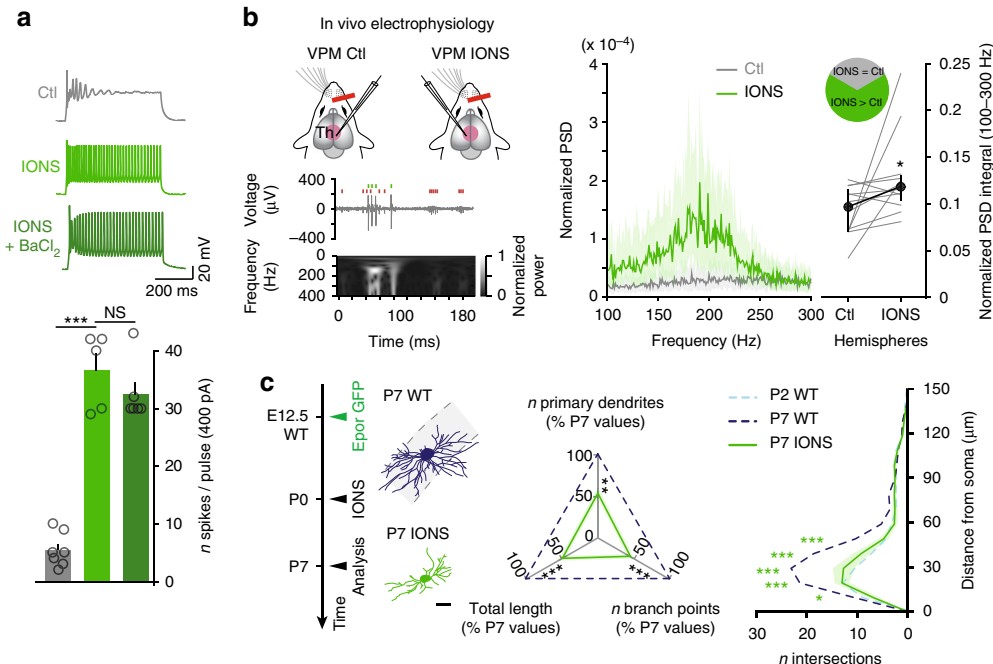

**Fig. 4** VPM neurons are hyperexcitable following neonatal lesions of whisker input pathways. **a** At P15, IONS leads to increased excitability of VPM neurons, which occludes the effects of the K+ channel blocker BaCl2. (Ctl $n = 7$, IONS $n = 5$, IONS + BaCl2 $n = 6$). One-way ANOVA with Tukey's post-hoc test, ***$P < 0.001$, NS, not significant. **b** IONS causes increased burstiness of VPM neurons in vivo. Left, top: multiunit activity recorded in VPM IONS. Two putative single units can be identified (green and red tick marks). Left, bottom: increased power in the 100–300 Hz frequency band during bursting activity. Center: sample traces showing VPM neuron activity in vivo. Shaded areas represent the interquartile range of the power spectral density (PSD). Right: summary plot of the normalized PSD integral. Lines represent individual mice ($n = 12$). Wilcoxon signed-rank test, *$P < 0.05$. **c** Dendritic complexity is strongly decreased following IONS (P7 IONS $n = 10$ from 2 mice). P7 WT and P2 WT data reported from Fig. 1b. One-way ANOVA with Tukey's post-hoc test for all statistical tests relating to dendritic complexity, except for Sholl analyses for which a two-way ANOVA with Tukey's post-hoc test was used. *$P < 0.05$, **$P < 0.01$, ***$P < 0.001$. Scale bar: 20 μm

We tested this model by performing IONS at birth to disrupt peripheral input to VPM neurons (VPM_IONS) and examined how input deprivation affects neuronal excitability. As was the case with ThGrin1KO neurons, VPM_IONS neurons displayed an increase in spiking at P15 when challenged with depolarizing current injections, which was occluded by blockade of K+ channels (Fig. 4a). As was the case in ThGrin1KO neurons, other electrophysiological parameters were not systematically affected by IONS (Supplementary Table 2). Neuronal hyperexcitability was already present at P7, in line with the early downregulation of Kv1.1 (Fig. 2b) and the developmental dynamics of the dendritic changes reported above (Supplementary Fig. 7 and Supplementary Table 3). Consistent with these findings, in vivo extracellular recordings at P20–23 revealed an increase in spike burstiness in VPM following IONS (Fig. 4b and Supplementary Fig. 8). As was the case in Grin1KO^ThEpor VPM neurons, dendritic complexity was strongly decreased in VPM_IONS neurons (Fig. 4c, and Supplementary Figs. 1f and 2). Thus, peripheral input ablation does not lead to VPM neuron silencing, but instead causes a (paradoxical) increase in neuronal excitability through decreased K+ conductances.

**Kv1.1 restores impaired maturation in VPM neurons.** If increased membrane excitability is responsible for the mis-maturation of Grin1KO^ThEpor VPM neurons and VPM_IONS neurons, then overexpression of Kcna1, by restoring neuronal excitability, should also restore dendritic maturation. Confirming this possibility, Kcna1 electroporation restored dendritic development in Grin1KO^ThEpor neurons (Fig. 5a, and Supplementary

Figs. 1g and 2). Similarly, overexpression of Kcna1, which decreased neuronal excitability to control levels (Supplementary Fig. 7 and Supplementary Table 3), was sufficient to restore dendritic maturation despite lack of whisker input in VPM_IONS neurons (Fig. 5b, and Supplementary Figs. 1h and 2). Together, these findings indicate that neuronal hyperexcitability is a critical determinant of abnormal dendritic development following genetic or surgical ablation of periphery-derived signals (Fig. 5c).

## Discussion

Our findings unveil a functional pathway linking input-dependent DNA epigenetic modifications, neuronal excitability, and dendritic maturation during early postnatal development. We propose that during postnatal development, as peripheral sensory input increases, postsynaptic activation of NMDARs induces Kv1.1 expression, which homeostatically maintains VPM neuron responses within a physiological range. In contrast, when sensory input is lacking or impaired, Kv1.1 expression is decreased, leading to compensatory increased neuronal excitability. Although Kcna1 expression levels have not been examined directly in Grin1KO^ThEpor neurons, overexpression of Kv1.1 restores the dendritic complexity of these cells to normal levels (Fig. 5a), supporting a downregulation of this transcript, as occurs in ThGrin1KO cells (Fig. 2b). Although the current study focuses on Kv1.1, which has a striking developmental regulation (Fig. 2b), other K+ channels were decreased in VPM neurons in ThGrin1KO mice (e.g., Kcna2 and Kcnn1, see Supplementary Table 1), which may act combinatorially with Kv channels to regulate excitability and dendritic maturation. Epigenetic

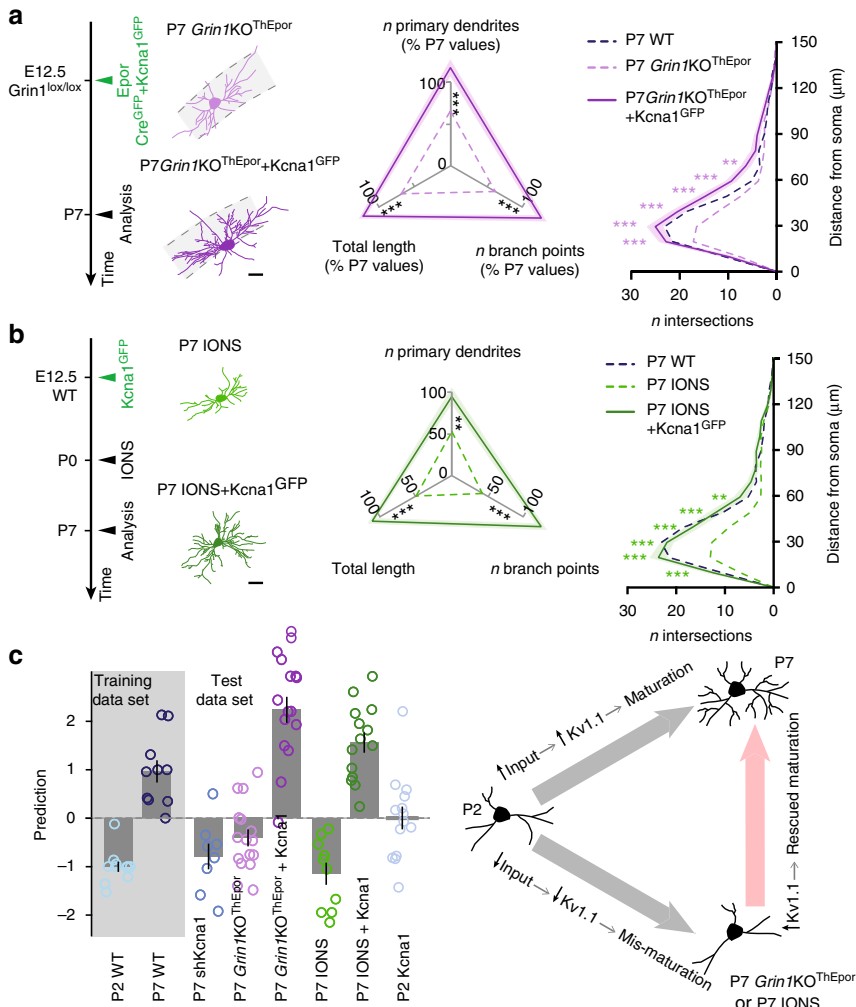

**Fig. 5** Overexpression of Kv1.1 enables normal dendritic development of *Grin1*KO[ThEpor] and VPM[IONS] neurons. **a** *Kcna1* electroporation enables dendritic development in *Grin1*KO[ThEpor] neurons (P7 *Grin1*KO[ThEpor] + Kcna1[GFP] $n = 15$ from 6 mice). P7 WT and P7 *Grin1*KO[ThEpor] data reported from Fig. 1b and d. Scale bar: 20 μm. **b** *Kcna1* overexpression enables the dendritic development of VPM[IONS] neurons (P7 IONS + Kcna1[GFP] $n = 14$ from 7 mice). P7 WT and P7 IONS data reported from Figs. 1b and 4c. Scale bar: 20 μm. **c** Left: linear model integrating the dendritic complexity parameters described above, using control P2 and P7 conditions as training data sets. Right: summary of the findings. One-way ANOVA with Tukey's post-hoc test for all statistical tests relating to dendritic complexity, except for Sholl analyses for which a two-way ANOVA with Tukey's post-hoc test was used. *$P < 0.05$, **$P < 0.01$, ***$P < 0.001$

regulation of this process has the advantage of allowing sustained transcriptional response even upon transient changes in the environment, thereby durably tuning neuronal excitability to fit external conditions as newborn mice start exploring their environment. Interestingly, GADD45B has also been associated with critical period plasticity in the visual cortex, where its expression is decreased by visual deprivation[28]. Therefore, regulation of GADD45B expression during critical periods could provide new strategies for increasing neuronal circuits plasticity in adult brain.

The persistence of barrels in *Gadd45b*[−/−] mice (Supplementary Fig. 6) and in a *Kcna1*[−/−] mouse[29] suggests that competitive interactions between neurons may be at play during barrel patterning. Indeed, mosaic deletion approaches as performed here do not necessarily reflect the situation found in whole body knockouts (see, e.g., Datwani et al.[20] vs. Mizuno et al.[16] for differences between global and mosaic manipulations of Grin1 in L4 barrel neurons). Indeed, the former approach introduces a competitive advantage/disadvantage for manipulated cells, which is not the

case in whole body knockouts, in which all cells are equally affected. Competition between manipulated and non-manipulated cells might be particularly relevant in studies of neuronal excitability, in which global neuronal silencing and local silencing have strikingly different effects (see, e.g., inter-ocular interactions during visual map development[30]; or effects of uni-lateral vs. bilateral silencing during callosal wiring[31]). In addition, regulatory mechanisms might come into play early in development to compensate for the missing gene in whole body knock-outs, as both Gadd45b and Kv1.1 each belong to a family of structurally overlapping proteins.

Although VPM barreloid neurons and L4 barrel neurons belong to the same functional pathway, NMDAR activation appears to have a different effect on dendritic growth in these two cell types: loss of Grin1 increases dendritic length in L4 neurons[20] but decreases it in VPM neurons (current study). These suggest that NMDARs can act on a variety of cellular differentiation programs and modulate features of neuronal development and circuit formation in a cell type-specific manner.

Although our study focuses on cell intrinsic electrophysiological changes in VPM neurons, synaptic input to these cells is likely also affected. Mosaic deletion of Grin1 in VPM neurons leads to decreased pruning of afferent PrV axons, yet most of these inputs appear to be non-functional, as upregulation of AMPARs was disrupted at these synapses[8]. Our findings are compatible with such a scenario, in that increased excitability could reflect a homeostatic "denervation hypersensitivity" following loss of input[32]. Similarly, changes in input to VPM neurons may also affect "top–down" afferents originating in the cortex. Although only little is known on the plasticity of these inputs following peripheral lesions, L5B input, which normally targets higher-order nuclei is functionally[18] and anatomically[33] rewired onto visual thalamic neurons following enucleation, resulting in a new genetic identity of these cells[18]. Whether a similar process is at play in the somatosensory system remains to be tested.

Neuronal hyperexcitability following IONS is somewhat surprising and may reflect denervation hypersensitivity secondary to loss of a critical input, as discussed above[32]. An important consequence of this latter finding is that developmental lesions to whisker input pathways are not equivalent as the "silencing" downstream target neurons, as in fact such procedures result in hyperactive neurons. Similar processes may occur along other sensory pathways; we have for instance shown that assembly of inhibitory circuits within the dorsolateral geniculate nucleus, which relays input from the retina, is disrupted following developmental enucleation[34]. Interestingly, secondary neuronal hyperactivity could be involved in maladaptive plasticity processes occurring following peripheral injuries, such as phantom limb pain[35].

Our results reveal that membrane hyperexcitability is a critical component of abnormal dendritic development following lesions to peripheral pathways and reducing neuronal excitability is sufficient to prevent the decrease in dendritic complexity observed following peripheral lesions. Importantly, although the mechanisms underlying hyperexcitability following genetic or surgical input manipulation may be distinct, they both result in impaired dendritic maturation and converge in being both restored by hyperpolarization. Given the tight link between input loss, hyperexcitability, and dendritic maturation reported here, it would be interesting to investigate neuronal silencing approaches to promote recovery of function following lesions to the central or peripheral nervous system.

## Methods

**Mice.** C57Bl/6 male and female pups and adult mice were used. Transgenic mice consist in Sert$^{Cre}$ (B6.129(Cg)-Slc6a4tm1(cre)Xz/J, The Jackson Laboratory, stock number 014554)[36], Grin1$^{flox}$ (B6.129S4-Grin1$^{tm2Stl}$/J, The Jackson Laboratory, stock number 005246), Ai14 transgenic reporter mice (B6.Cg-Gt(ROSA)26Sor$^{tm14(CAG-tdTomato)Hze}$/J, The Jackson Laboratory, stock number 007914), Gadd45b$^{-/-}$ (B6;129S6-Gadd45b$^{tm1Flv}$/J, The Jackson Laboratory, stock number 013101), and Kcna1$^{-/-}$ (gift from Bruce Tempel)[29] were used for all studies. All experimental procedures were approved by the Geneva Cantonal Veterinary Authority.

**Generation and genotyping of mice.** ThGrin1KO were obtained by crossing Sert$^{Cre/+}$Grin1$^{flox/+}$ mice with Grin1$^{flox/flox}$ mice. Genotypes of mice were determined by PCR analysis of genomic DNA prepared from tail. PCR primer sets for Sert$^{Cre}$ were: 5′-ATT TGC CTG CAT TAC CGG TCG-3′ and 5′-CCC CAG AAA TGC CAG ATT ACG TAT ATC-3′. PCR primer sets for Grin1$^{flox}$ were: 5′-GTG AGC TGC ACT TCC AGA AG-3′, 5′-GAC TTT CGG CAT GTG AAA TG-3′, 5′-CTT GGG TGG AGA GGC TAT TC-3′, and 5′-AGG TGA GAT GAC AGG AGA TC-3′. PCR primer sets for Gadd45b$^{-/-}$ were 5′-GCA ACC CCA GTA ACT TTG GA-3′, 5′-CCT GCA GGA GAG AAG GAG TG-3′, and 5′-CTT CCA TTT GTC ACG TCC TG-3′.

**ION section.** An IONS was performed on P0 pups[37]. Animals were anesthetized by hypothermia. A unilateral skin incision was made between the eye and the whisker pad, and the infraorbital nerve, which innervates the whisker pad, was carefully cut

with sterile microscissors. The pups were allowed to recover on a heating pad before being returned to their mother.

**Tissue microdissection, microarray and qPCR.** Microdissection of VPM from one litter corresponds to one biological replicate ($n = 3$ for each condition). For ThGrin1KO microarrays, collection of samples was done at P1 and WT littermates were used as controls. For IONS microarrays, collection of the samples was done at P3 and the VPM ipsilateral to the IONS was used as control. Fresh coronal brain sections (140 µm) were cut on a vibrating microtome (Leica, VT1000S) and thalamic nuclei were visually identified and microdissected using a Leica Dissecting Microscope (Leica, M165FC) in ice-cold oxygenated artificial cerebrospinal fluid under RNAse free conditions. Samples were stored in RNAlater (Sigma) at −80 °C.

For microarrays, RNA was extracted using an RNeasy kit (Qiagen) and two-cycle amplification and labeling were performed according to Affymetrix protocols using Superscript complementary DNA synthesis kit (Invitrogen), MEGAscript T7 kit and MessageAmp II aRNA amplification kit (Ambion). Experiments were performed blindly. Labeled cRNA was fragmented and hybridized to Affymetrix Mouse Genome 430 2.0 Array. GeneChips were incubated at 45 °C for 16 h with biotin-labeled cRNA probes, and then washed and stained using a streptavidin–phycoerythrin conjugate with antibody amplification as described in Affymetrix protocol, using Affymetrix GeneChip Fluidics Station 450. GeneChips were scanned on a GCS3000 scanner (Affymetrix). Microarray CEL files were read and normalized using 'affy' and 'gcrma' R packages, and transformed in log2. Data were analyzed as previously described[18]. Some of these arrays were used in ref.[17].

For quantitative PCR, cDNA was synthesized from 1 µg of cRNA (from the second amplification) using a mix of random hexamers—oligo d(T) primers and PrimerScript reverse transcriptase enzyme (Takara Bio Inc. Kit) following suppliers instructions. SYBR Green assays were designed using the program Primer Express v 2.0 (Applied Biosystems) with default parameters. Amplicon sequences were aligned against the mouse genome by BLAST, to ensure that they were specific for the gene being tested. Oligonucleotides were obtained from Invitrogen. The efficiency of each design was tested with serial dilutions of cDNA. PCR reactions (10 µl volume) contained diluted cDNA, 2 x Power SYBR Green Master Mix (Applied Biosystems), and 300 nM of forward and reverse primers. PCR were performed on a SDS 7900 HT instrument (Applied Biosystems) with the following parameters: 50 °C for 2 min, 95 °C for 10 min, and 45 cycles of 95 °C for 15 s/60 °C for 1 min. Each reaction was performed in three replicates on 384-well plates. Raw Ct values were obtained with SDS 2.2 (Applied Biosystems) and imported in Excel. Normalisation factor and fold changes were calculated using the GeNorm method[38].

**In utero electroporation.** Timed pregnant C57Bl/6 or Grin1$^{flox/flox}$ mice with E12.5 embryos were anesthetized with isoflurane (4.5% induction, 2.5% during the surgery) and uterine horns were successively exposed after a midline laparotomy. Embryos were injected with 200 nL plasmid DNA solution (prepared in 0.9% NaCl, 0.3 mg ml$^{-1}$ Fast Green) into the third ventricle through the uterine wall. pCAG-IRES-GFP (1 µg µl$^{-1}$, gift from Guillermina López-Bendito) was injected alone or co-electroporated, with a Kcna1 shRNA (2 µg µl$^{-1}$). Small hairpin (sh) RNA was purchased from Thermo Scientific (TRC Mouse Kcna1 shRNA, RMM4534-EG16485). pCAG-Cre:GFP was a gift from Connie Cepko[39] (Addgene, plasmid 13776) and was injected in Grin1$^{flox/flox}$ embryos at 2 µg µl$^{-1}$. pCAG-KCNA1-IRES-GFP was constructed by inserting the human Kcna1 sequence (Dharmacon, MHS6278-202857646) into the pCAG-IRES-GFP. pCAG-KCNA1-IRES-GFP was injected at 2 µg µl$^{-1}$ for Kcna1 overexpression experiments. Embryos were electroporated by holding their head between tweezers-style circular electrodes (3 mm diameter, Sonidel Limited, UK) across the uterus wall, whereas five pulses (35 V, 50 ms duration with 950 ms intervals) were delivered with a square-wave electroporator (Nepa Gene, Sonidel Limited). The uterine horns were returned into the abdominal cavity, the wall and skin were sutured, and the embryos were allowed to continue their normal development until P2 or P7.

**Histology.** Postnatal mice were perfused with 4% paraformaldehyde (PFA) and brains were fixed overnight in 4% PFA at 4 °C. Fifty-micrometer vibratome sections (Leica, VT1000S) were used for all histologic experiments.

ISH on slides was performed according to methods described previously[19]. For antisense probe synthesis, RNA extraction was performed with RNeasy mini kit (Qiagen). Following RNA extraction, cDNA was generated using SuperScript III kit (Invitrogen). The desired cDNA sequences were amplified by PCR using specific primers. PCR products were used to generate DIG-labeled antisense RNA probes (Roche) using T7 RNA polymerase (Roche). ISH was performed on slides, on 50 µm RNAse-free brain sections. Briefly, hybridization was carried out overnight at 60 °C with the DIG-labeled RNA probes. Following hybridization, sections were washed and incubated with alkaline phosphatase-conjugated antidigoxigenin antibody (1:2,000; Roche) overnight at 4 °C. Following incubation, sections were washed and the color reaction was carried out overnight at 4 °C in a solution containing NBT/BCIP (Roche). After color revelation, sections were washed, post-fixed for 30 min in 4% PFA and mounted with Fluoromount (Sigma).

For fluorescence immunohistochemistry, brain sections were incubated 1 h at room temperature in a blocking solution containing 3% bovine serum albumin and 0.3% Triton X-100 in PBS, and incubated overnight at 4 °C with primary antibodies: guinea pig anti-VGLUT2 (1:2,000; Millipore, AB2251) and rabbit anti-GFP (1:1,000; Invitrogen, A11122). Sections were rinsed three times in PBS and incubated 1 h at room temperature with the corresponding secondary antibodies (1:500).

For cytochrome oxidase staining, free-floating sections were placed in a solution of 0.5 mg ml$^{-1}$ DAB, 0.5 mg ml$^{-1}$ Cytochrome C (Sigma), 40 mg ml$^{-1}$ sucrose, 0.1 mM Tris pH 7.6 at 37 °C until staining appeared.

**Imaging and quantification**. All images were acquired on an Eclipse 90i fluorescence microscope (Nikon, Japan) or on a Zeiss LSM 700 Live confocal system (Carl Zeiss). Morphological quantifications were done blindly with respect to experimental conditions. Only one criterion was used to select cells for quantification: no overlap with other labelled neurons; specifically, no overlap across primary dendrites.

Control P2 ($n = 11$ cells from 2 mice), control P7 ($n = 10$ cells from 2 mice), P7 Grin1KO$^{ThEpor}$ ($n = 16$ cells from 3 mice), P7 Grin1KO$^{ThEpor}$ + Kcna1$^{GFP}$ ($n = 15$ cells from 6 mice), P7 shKcna1 ($n = 8$ cells from 2 mice), P7 shKcna1 + Kcna1$^{GFP}$ ($n = 13$ cells from 4 mice), P2 Kcna1$^{GFP}$ ($n = 14$ cells from 4 mice), P7 IONS ($n = 10$ cells from 2 mice), and P7 IONS + Kcna1$^{GFP}$ ($n = 14$ cells from 7 mice) VPM neurons were reconstructed for quantitative analysis of neuronal dendritic arborization. In order to quantify changes in dendritic arborization, we quantified the number of primary dendrite, number of branch point and total length. All values were normalized and expressed as a percentage of control P7 VPM neurons values. Data are expressed as mean ± SEM and statistical analysis was performed with one-way analysis of variance (ANOVA) with Tukey's post-hoc test with significance level set at $P < 0.05$. Values for Fig. 1b are as follows: P2 WT number of primary dendrites 58.5% ± 4.71, P2 WT number of branch points 40.06% ± 3.25, P2 WT total dendritic length 48.03% ± 4.17, P7 WT number of primary dendrites 100% ± 6.32, P7 WT number of branch points 100% ± 11.18, and P7 WT total dendritic length 100% ± 8.14. Values for Fig. 1d are as follows: P7 Grin1KO$^{ThEpor}$ number of primary dendrites 65.28% ± 5.51, P7 Grin1KO$^{ThEpor}$ number of branch points 63.39% ± 3.81, and P7 Grin1KO$^{ThEpor}$ total dendritic length 69.63% ± 4.32. Values in Fig. 2d are as follows: P7 shKcna1 number of primary dendrites 53.1% ± 6.43, P7 shKcna1 number of branch points 56.08% ± 4.61, and P7 shKcna1 total dendritic length 59.58% ± 4.41. Values in Fig. 2e are as follows: P2 + Kcna1 number of primary dendrites 89.42% ± 5.74, P2 + Kcna1 number of branch points 64.32% ± 6.2, and P2 + Kcna1 total dendritic length 67.94% ± 5.26. Values in Fig. 4c are as follows: P7 IONS number of primary dendrites 52.87% ± 5.97, P7 IONS number of branch points 45.51% ± 4.91, and P7 IONS total dendritic length 50.88% ± 5.53. Values in Fig. 5a are as follows: P7 Grin1KO$^{ThEpor}$ + Kcna1 number of primary dendrites 117.24% ± 10, P7 Grin1KO$^{ThEpor}$ + Kcna1 number of branch points 125.75% ± 7.08, and P7 Grin1KO$^{ThEpor}$ + Kcna1 total dendritic length 120.53% ± 5.66. Values in Fig. 5b are as follows: P7 IONS + Kcna1 number of primary dendrites 94.25% ± 9.42, P7 IONS + Kcna1 number of branch points 123.97% ± 7.46, and P7 IONS + Kcna1 total dendritic length 109.29% ± 5.97. Values in Supplementary Fig. 4a are as follows: P7 shKcna1 + Kcna1 number of primary dendrites 80.45% ± 6.43, P7 shKcna1 + Kcna1 number of branch points 121.55% ± 6.9, and P7 shKcna1 + Kcna1 total dendritic length 104.64% ± 6.9.

Sholl analysis was used to study dendritic branching. Using the Image J Sholl Analysis plug-in ref. [40], the number of dendrite crossing each 10 μm radius ring progressively more distal to the soma was counted in each condition. Data are expressed as mean ± S.E.M. and statistical comparisons were done using two-way ANOVA with Tukey's post-hoc test with significance level set at $P < 0.05$. In Figs. 1e, 2d, e, 4c, and 5a, b, and Supplementary Fig. 4 values for the Scholl analysis have been reported to facilitate the interpretation of the new data.

For Fig. 5c, we trained a linear nu-support vector machine integrating the dendritic complexity parameters described above (i.e., the number of primary dendrite, number of branch point, total length and Sholl analysis), using control P2 and P7 conditions as training data sets[41, 42]. The data were scaled and centered and the SVM model was constructed using the Support Vector Machine package "E1071" available on R using as type parameter "nu-classification" and nu parameter "0.5". The other conditions were then predicted using this model and associated to a P2 sample if the prediction value was < 0 or to a P7 sample if the prediction value was > 0.

**In vitro electrophysiology**. Mice were deeply anesthetized with isoflurane and were then decapitated. Brains were cut in cooled and oxygenated (95% O$_2$ and 5% CO$_2$) artificial cerebrospinal fluid containing: 119 mM NaCl, 2.5 mM KCl, 1.3 mM MgCl, 2.5 mM CaCl$_2$, 1.0 mM Na$_2$HPO$_4$, 26.2 mM NaHCO$_3$, and 11 mM glucose. Coronal slices containing the thalamus were kept at room temperature and were allowed to recover for at least 1 h before recording. The internal solution contained 140 mM potassium gluconate, 5 mM KCl, 10 mM HEPES, 0.2 mM EGTA, 2 mM MgCl$_2$, 4 mM Na$_2$ATP, 0.3 mM Na$_3$GTP, and 10 mM sodium creatine phosphate. Currents were amplified (Multiclamp 700B, Axon Instruments), filtered at 5 kHz, and digitized at 20 kHz (National Instruments Board PCI-MIO-16E4, Igor, WaveMetrics). The liquid junction potential was + 12 mV. Spike measurements for a given cell are the mean values measured from one to three cycles of current steps

(500 ms duration at 0.1 Hz, 0 to + 400 pA range with a 50 pA step increment). For Figs. 2c and 4a, ~ P15 control, ThGrin1KO and IONS mice were used. The K$^+$ channel blocker BaCl$_2$ (1 mM, Sigma-Aldrich) was bath-applied. For Fig. 3d, P5 control and Gadd45b$^{-/-}$ mice were used.

For Supplementary Fig. 6a, P7 control, IONS, and IONS + Kcna1$^{GFP}$ mice were used. Data are expressed as mean ± SEM and statistical significance was determined by one-way ANOVA with Tukey's post-hoc test with significance level set at $P < 0.05$.

**In vivo extracellular electrophysiology**. IONS mice ($n = 12$, ~ P20–23) and control mice ($n = 4$, ~ P20–21) of both sexes were used to obtain in vivo extracellular recordings. The animals were pretreated with glycopyrrolate (Robinul, 0.01 mg kg$^{-1}$, subcutaneously) at least 15 min before anesthesia, in order to prevent bradycardia and excessive salivation. Then, they were anesthetized with a mixture of medetomidin (Dorbene, 0.1 mg kg$^{-1}$), midazolam (Dormicum, 2.5 mg kg$^{-1}$), and fentanyl (Duragesic, 0.025 mg kg$^{-1}$) in sterile NaCl 0.9%. Their head was shaved and depilatory cream (Veet) was applied. An incision was made to the skin and the skull was exposed. Lidocaine 1% was applied on the wound edges. Dental cement (Lang) was applied around the bregmatic bones to create a "bath." The head was aligned using bregma and lambda as reference points and a hole was drilled above the VPM under stereotaxic guidance using a pneumatic dental drill (coordinates relative to bregma: AP: − 1.5, ML: 1.8). Signals were recorded using borosilicate glass pipettes (Science Products) containing 2 M NaCl ( < 1 MΩ). Chicago Sky Blue (0.2% v/v) was added in the pipette in a subset of recordings in order to mark the recording area. The pipettes were lowered into VPM (3.1 mm) of both hemispheres and a reference electrode was placed in the bath containing artificial cerebrospinal fluid (125 mM NaCl, 5 mM KCl, 10 mM glucose, 10 mM HEPES, 2 mM CaCl$_2$, and 2 mM MgSO$_4$ in distilled H$_2$O). After a waiting period of 5–10 min, spontaneous activity was differentially recorded, filtered (0.1 Hz – 10 kHz) and amplified (2–5 K) with an extracellular amplifier (ER-1, Cygnus Technology), and acquired at 40 kHz through Igor Pro (WaveMetrics). Ten minutes of spontaneous activity were typically recorded from each hemisphere before moving to the other hemisphere. The sequence of recording from the two hemispheres was reversed between successive mice. Multi-whisker stimulation of the non-deprived whisker pad was performed in a subset of mice, in order to confirm the targeting of the recording pipette. Signal analysis was performed in MATLAB (MathWorks). The raw signals were divided into traces of 10 s. These traces were decimated to extract the local field potential by zero-phase, low pass (450 Hz) filtered, and downsampled to a sampling rate of 1 kHz. Multi-unit activity (MUA) was extracted from these traces by zero-phase, band-pass filtered (0.3–4.5 kHz). The resulting signal was rectified, low-pass filtered (300 Hz) with a Gaussian window, and downsampled to 1 kHz (decimated MUA). The power spectral density was calculated for these traces and was then normalized to the DC component. The time-frequency transform[43] of the decimated MUA traces was used for visualization. The raw traces were also band-passed, zero-phase filtered (0.3 - 6.7 kHz) and down-sampled to 20 kHz. Wave_clus v2.0[44] was used for automatic spike detection and sorting of single units from these signals. One to three units were identified per recording and their time stamps were extracted. Putative units that were found to spike at a rate of < 0.1 Hz on average were discarded. The squared coefficient of variation of the inter-spike interval distribution was used as a measure of burstiness of a single unit[45]. Statistical comparisons were done using Wilcoxon's signed-rank test with significance level set at $P < 0.05$.

**Pyrosequencing**. Microdissection of VPM from one litter (~ 8 pups) corresponds to one biological replicate ($n = 2$ for WT and $n = 3$ for Gadd45b$^{-/-}$). Collection of the samples was done at P3. Fresh coronal brain sections (140 μm) were cut on a vibrating microtome (Leica, VT1000S) and thalamic nuclei were visually identified and microdissected using a Leica Dissecting Microscope (Leica, M165FC) in icecold oxygenated artificial cerebrospinal fluid. DNA was extracted using a DNeasy Mini Kit (Qiagen). DNA has been bisulfite converted using the EZ DNA Methylation-Gold Kit (Zymo Research) according to Bibikova recommendations[46]. Briefly, ~ 200 ng of DNA was converted during 16 h (95 °C 10 min + (95 °C 30 s, 50 °C 60 min) x 16 + 4 °C), desulphonated, purified, and eluted with 30 μl of water. Due to the low amount of starting DNA a nested-PCR strategy was designed: first, PCR was performed using non-biotinylated primers (F: 5′-TTG AGG GTT TTT AGT GAA GTT T-3′; R: 5′-TCC CAT CAA AAT ACT TCT AAT TTT-3′) and PCR-cycled (95 °C 10 min + (95 °C 30 s, 60 °C 30 s, 72 s, 30 s) x 40 + 4 °C) using a HotstartTaq Plus polymerase (Qiagen), 2.5 mM Mg$^{2+}$ final concentration, and 2 μl of bisulfite converted DNA as template. For the second PCR, biotinylated primers were used (F: 5′-[BTN]-TGA AGT TTT TTT GTT ATT TTT GGG GTA TT-3′, R: 5′-TTT CTA CCC TCR TAT CAC TTT TTA T-3′) and PCR-amplified using similar conditions, and 2 μl of first PCR as template. The resulting DNA from second PCR was purified and single DNA strand captured using Streptavidin beads. The CUX1-binding site region was pyrosequenced (PSQ 96MA) using the reverse primer of the second primer as a sequencing primer. Average values for the fourCpGs investigated in Gadd45b promoter region are represented. Data are expressed as mean ± SEM and statistical comparisons were done using Kruskal–Wallis test with significance level set at $P < 0.05$.

**Statistical analysis**. No statistics were used to determine group sample size; however, sample sizes were similar to those used in previous publications from our group and others. No randomization was performed. Tests were performed assuming equal variances except when indicated. *n*-values refer to cell numbers unless specified otherwise. Values are represented as means ± SEM throughout the manuscript. *p*-values: NS, not significant ($P > 0.05$), *$P < 0.05$, **$P < 0.01$, ***$P < 0.001$.

**Data availability**. All relevant data are available from the authors.

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

## Acknowledgements

We thank A. Benoit and M. Lanzillo for technical assistance, and members of our laboratories for helpful discussions and suggestions. We wish to thank Bruce Tempel for the gift of Kcna1⁻/⁻ mouse, Guillermina López-Bendito for the gift of the pCAG-IRES-GFP, and Natalia Baumann for help with the analysis. Work in the Jabaudon laboratory is supported by the Swiss National Science Foundation (PP00P3_123447), the Synapsis Foundation, and the Brain and Behavior Foundation (NARSAD Grant). A.H. is supported by the Swiss National Science Foundation. J.G.'s laboratory is supported by the European Research Council (ERC-2015-StG 678832), by the Swiss National Science Foundation (31003A_155898), the National Competence Center for Research SYNAPSY, the SYNAPSIS Foundation, the Béatrice Ederer-Weber Stiftung, the Floshield Foundation, and the Alzheimer's Association (NIRG-15-363964). J.G. is an MQ fellow and a NARSAD Independent Investigator. Initial work in J.D.M.'s laboratory was supported by US National Institutes of Health grants (NS045523, NS041590, and NS075672) and the Pearlstein and Seidman Funds, and J.D.M. is an Allen Distinguished Investigator of the Paul G. Allen Frontiers Group.

## Author contributions

D.J. and J.D.M. conceived the initial project. L.F. designed the experimental layout and scientific model, and wrote the manuscript with help from D.J. and other authors. L.F., V. K., J.V.S.-M., S.F., K.K.-K., G.P., and C.B. performed experiments and analyses. L.T.

helped with data analyses. D.J., J.D.M., A.H., and J.G. advised on experiments and manuscript preparation.

## Additional information

**Competing interests:** The authors declare no competing financial interests.

