## [Peer Review File · Nature Communications]

Reviewers' expertise:

Reviewer #1: Dendrite morphogenesis, NMDAR;
Reviewer #2: Development of thalamocortical circuits;
Reviewer #3: Barrel cortex development and plasticity.

Reviewers' comments:

Reviewer #1 (Remarks to the Author):

In this paper, Frangeul et al asked how the dendrite maturation of VPM neurons in neonatal stage is regulated by neural activity. By analyzing thalamus-specific NMDAR knockout mice and IONS mice, they showed that dendritic maturation of VPM neurons could be regulated by membrane excitability. They also showed that membrane excitability can be regulated by NMDAR-mediated Kv1.1 expression via GADD45B. Most strikingly, morphological abnormalities of VPM neuron dendrites induced by NMDAR knockout and IONS were rescued by Kv1.1 overexpression. Based on these results, the authors claimed that neural activity facilitates dendritic maturation by maintaining the neuronal excitability within a physiological range. Understanding the mechanisms of activity-dependent neural circuit formation is a central theme in neuroscience and mouse somatosensory system is an ideal model for the purpose. Their hypothesis is novel and extremely interesting. Although this paper is potentially important, I have several concerns as detailed below.

Major concerns:

1. The authors claimed that abnormal VPM neuron dendritic morphology in ThGrin1KO and IONS mice is due to hyper-excitability of these neurons. The first postnatal week is critical for VPM neuron dendrite maturation, and indeed they analyzed dendrite morphology at P2 and P7. On the other hand, for the neuron excitability, they used P15 and P22 mice for in vitro and in vivo analyses, respectively. It is well known that IONS (Grin1KO, too) rearranges neural circuits drastically during postnatal development probably by compensatory mechanisms (e.g. Marques-Smith et al., Neuron 89, 536-549, 2016). Therefore, it is very likely that neuronal excitability is totally different between P7 and P15 (or P22). The authors should analyze neural excitability during the first postnatal week such as at P7. If VPM recording in young mice in vivo is technically difficult, using rats can be considered.
2. For both dendrite morphology analyses and physiological analyses, it appears that the authors used the same controls many times. Did the authors use statistics for "multiple comparison"? Otherwise, it is not appropriate. If they used multiple comparison, the statistic method they used should be stated clearly.
3. Even if the authors used multiple comparison statistics properly, I still have a concern. Because in utero electroporation at E12.5 is technically difficult, it is not easy to label neurons consistently. If a neuron is brighter than another neuron, even if both neurons have

the same dendritic length, dendritic length of the former could be estimated longer. Also because the morphologies of individual neuron dendrites are highly variable, it is very important to avoid bias in selecting neurons for analyses. A useful way to solve these problems would be to prepare distinct control for each analysis and conduct experiments in a blind fashion. I do not think that the authors can repeat all the experiments again with this way. But at least, some most critical experiments should be repeated for a new sample/control set in a blind fashion to confirm their conclusions.

4. It is generally thought that the NMDARs regulate circuit refinement (including dendritic maturation) via Hebbian-type mechanism. While, the authors claimed that NMDAR's role is just to suppress the membrane excitability and simple overexpression of Kir1.1 can rescue the NMDAR knockout phenotype. If it is the case, it would be very surprising. A problem of their analysis is that their conclusions rely on very simple parameters of dendritic morphology. They used only total dendrite length, number of primary dendrites, number of branch points, and Sholl analyses. Importance of dendritic morphology of VPM neurons is not just complexity. A more important aspect is whether dendrites confined to a single barreloid or not. The authors should analyze dendritic expansion in respect to barreloid patterns. Although ThGrin1KO and IONS mice do not have barreloids, in utero electroporation-based knockdown (knockout) and rescue could be possible.

5. (Figure 1a, Supplementary Figure 1) Please show better images for GFP-labeled neurons. With these images, dendrite morphologies are not very clear. In addition, barreloid pattern should be shown along with GFP-labeled dendrites.

Minor comments:

1. In figures and/or figure legends, mouse age and sample numbers (both mouse number and cell number) used should be stated clearly. In particular, mouse age information for electrophysiology data is necessary in figures and figure legends.

2. (Figure 1) Please show an image of P2 GFP-labeled dendrites.

3. (Figure 1a right) What are white dotted lines?

4. (Figure 1a left) Brainstem-thalamus axons do not cross the midline. In this figure, whiskers should be positioned on the left side.

5. (Figure 1 legend) "(d) Dendritic maturation is DELAYED ----" How can the authors use the word, "DELAYED"? Without analyzing other ages such as at P2 and in adulthood, the authors cannot claim that the dendritic maturation is "DELAYED". Similarly, in text, avoid using "delayed (e.g. Main text p.5)" and "precocious development (Main text p.6)"

6. (Figure 2a, 2b,) What are red arrowheads?

7. (Figure 2) Mouse age and sample numbers should be clearly stated in the legend.

8. (Figure 3c) What are arrowheads? Describe the mouse number used for in situ (Fig.2, too).

9. (Supplementary Figure 1) What are red dotted lines? Please specify which traces are corresponding to the GFP images.
10. (Supplementary Figure 3b) Expression outside the VB looks higher in TgGrin1KO than in WT. Why?
11. (Methods) There are multiple lines of Sert Cre mice. Please specify mouse line that you used.
12. (Method) What is pCAG-IRES-GFP? Please describe how the authors constructed or obtained this plasmid.
13. (Method) (Imaging and quantification) Please describe mouse number used for analyses.

Reviewer #2 (Remarks to the Author):

In this manuscript, Frangeul et al. uses mouse somatosensory system to demonstrate that removing peripheral sensory input or deleting a subunit of NMDAR in thalamic projection neurons causes increased excitability and impaired maturation of dendritic morphology of these neurons and defective whisker-specific patterning of the thalamic nucleus. They further show that increased excitability is due to decreased expression of the potassium channel Kv1.1 and that *Kcna1* expression is epigenetically regulated by GADD45B, which is induced by NMDAR activation. These findings are novel and will be of great interest to the broad audience of developmental neurobiology.

To support the above conclusions, the authors used a wide variety of approaches including mouse genetics, in utero electroporation, analysis of gene expression, neuronal morphology and electrophysiology. Overall, the data are clearly presented and support their conclusions. In addition, experimental procedures are detailed enough and statistical analysis are appropriate. However, there are some missing experimental data or discussion, which makes it harder to think about this paper in the context of previously published work.

First, the authors show that barreloid formation in VPM is disrupted in *ThGrin1KO* mice as well as *IONS* mice, which has been published by others. However, whether the *Gadd45b* knockout or *Kcna1* knockdown mice have similar defects is not addressed. If the GADD45B-Kv1.1 pathway is an essential downstream pathway of NMDAR and is central to the role of NMDAR or peripheral input in the formation of whisker-specific patterning in the somatosensory system, it should be expected that *Gadd45b* knockout and *Kcna1* knockdown mice also show similar barreloid phenotypes as *ThGrin1KO* or *INONS* mice. Authors should show and discuss these data.

Second, the Zhang et al. (2014) paper used a similar strategy to knockout *Grin1* in thalamic neurons and found that pruning and strengthening of immature synapses are blocked in neurons without NMDARs. In addition, Lee et al., (2005) found that in *Grin1* KD mice, VPM neurons had longer dendrites with no orientation preference. How do these results compare with the findings in the current manuscript in which dendritic arborization is reduced and

excitability is increased when either of the molecular pathway components is disrupted? It will be informative and feasible to analyze the symmetry of dendrites shown in Supplemental Fig.1.

Reviewer #3 (Remarks to the Author):

This study by Frangeul et al. is an elegant investigation of the mechanisms involved in the control of dendritic patterning in mouse somatosensory thalamus during development. The authors demonstrate the NMDA receptor dependence and interaction with KV1.1 that underlies input-dependent maturation and excitability of VPM neurons. The strength of the paper is the identification of epigenetic regulation of this process. The electrophysiological characterization of the excitability changes and relation to dendritic structure is very important as well, but lacking in thoroughness on the biophysical level. My critique relates primarily to the latter shortcomings, as well as places where the methods presented are insufficient for other researchers to replicate these findings, and finally issues that should be clarified with text revision.

Major.

1. The in vitro electrophysiological properties are woefully under-analyzed. In addition to the spike number measurements that the authors present, a number of important properties should be described. These include (but are not necessarily restricted to) resting membrane potential, input resistance, series resistance, action potential threshold (rheobase), action potential height and width.

2. The authors found that decreased branching is associated with hyper-excitability. With some manipulations, increased branching was found (shRNA; Kcna1 electroporation). If the authors want to solidify the relationship between branching and excitability, experiments should be performed in the neurons with increased branching.

In the Kcna1 electroporation experiment (Fig. 5), it appears the authors did not test the electrophysiological properties of the neurons and the conclusions are based on dendritic branching alone. Conclusions about the excitability of the Kcna1 electroporated neurons seem unsupported.

3. While the authors use a nerve lesion experiment to show a relationship between sensory input and dendritic maturation in thalamic neurons, there is nothing done to measure properties of the synaptic input in a direct way, i.e., via stimulation of axons in slice recordings. Therefore, there is no information about how the branching patterns and excitability relate to properties of altered synaptic input. It is possible that the important changes in burstiness found in vivo could be due to presynaptic hyper-excitability (i.e. via increased excitability in trigeminal ganglion). It would be ideal if the authors could perform experiments to measure the properties of synaptic input to thalamic neurons to test whether they have changed. But at minimum, this topic needs to be thoroughly discussed as to whether the changes in branching and excitability measured in this study can be attributed to thalamic neurons themselves, or to presynaptic inputs from trigeminal ganglion or top-down inputs.

4. The Methods contain insufficient information for others to replicate the experiments. A. In the ION section description, the authors state that it was performed 'as previously described', yet in the cited study, the methods do not describe the procedure, they only reference a paper from 1990. This is an inappropriate self-citation that contains no information and should be removed. The ION procedure should be described in the current paper, as differences are bound to exist compared to a paper from a different lab from 27 years ago. B. In the Imaging and quantification part, a link or reference should be provided for the Image J Sholl Analysis pug-in. C. In the next paragraph, good luck to any researcher trying to replicate the SVM used here. The sentence 'The SVM model was constructed using dedicated R packages' leaves the reader with essentially or useful information about the SVM model or how it was constructed. More information needs to be provided.

Minor

1. In the experiments where electroporation is used to express Cre-GFP in Grin1-lox/lox mice (e.g., first part of Results), it should be stated that thalamic neurons are not the only target of Grin1 deletion. The authors should state any potential issues related to upstream/downstream effects from potential indirect effects on thalamic neurons due to global deletion of Grin1. Specifically, the sentence '... genetically ablated these receptors in VPM neurons...' needs to state that it was not exclusively in these neurons.

2. The word 'rescue' in the abstract implies that the authors performed experiments to alter dendritic maturation after it was altered by sensory deprivation. This is not true because the genetic manipulation was performed before sectioning the nerve. The word 'enable' (or alternative) may better capture the nature of the experiment.

3. I suggest changing the term 'proof-of-principle' in the last paragraph of Introduction. To my ear this sounds like the study was done only on a few neurons to show the results 'in principle', as one might do in a methods paper. Consider 'model system' or alternative.

4. Results, first paragraph: The sentence 'NMDARs have been involved in..' Involved should be changed to 'implicated in' or alternative.

5. The authors identify Kv1.1 as a key target of the changes in excitability, but there is no discussion of changes in expression of other ion channels that could contribute to this phenomenon. The issue of whether Kv1.1 fully explains the effects should be addressed in the Discussion.

6. The meaning of the penultimate sentence in the Discussion is unclear ('... regulated according to context.').

Frangéul et al., Answer to Reviewers' Comments

Reviewer #1

In this paper, Frangéul et al asked how the dendrite maturation of VPM neurons in neonatal stage is regulated by neural activity. By analyzing thalamus-specific NMDAR knockout mice and IONS mice, they showed that dendritic maturation of VPM neurons could be regulated by membrane excitability. They also showed that membrane excitability can be regulated by NMDAR-mediated Kv1.1 expression via GADD45B. Most strikingly, morphological abnormalities of VPM neuron dendrites induced by NMDAR knockout and IONS were rescued by Kv1.1 overexpression. Based on these results, the authors claimed that neural activity facilitates dendritic maturation by maintaining the neuronal excitability within a physiological range. Understanding the mechanisms of activity-dependent neural circuit formation is a central theme in neuroscience and mouse somatosensory system is an ideal model for the purpose. Their hypothesis is novel and extremely interesting. Although this paper is potentially important, I have several concerns as detailed bellow.

We thank the reviewer for these comments and provide answers to the points raised below.

Major concerns:

1. The authors claimed that abnormal VPM neuron dendritic morphology in ThGrin1KO and IONS mice is due to hyper-excitability of these neurons. The first postnatal week is critical for VPM neuron dendrite maturation, and indeed they analyzed dendrite morphology at P2 and P7. On the other hand, for the neuron excitability, they used P15 and P22 mice for in vitro and in vivo analyses, respectively. It is well known that IONS (Grin1KO, too) rearranges neural circuits drastically during postnatal development probably by compensatory mechanisms (e.g. Marques-Smith et al., *Neuron* 89, 536-549, 2016). Therefore, it is very likely that neuronal excitability is totally different between P7 and P15 (or P22). The authors should analyze neural excitability during the first postnatal week such as at P7. If VPM recording in young mice in vivo is technically difficult, using rats can be considered.

We have now performed P7 slice recordings and compared them to our existing P15 data as requested. **Our results show that neuronal hyperexcitability is already present at P7, which is consistent with the early downregulation of Kv1.1 expression we report here (see Fig. 2b). These new data are now presented in Supplementary Fig. 6a of the main manuscript, and also shown below.**

Following the request of Reviewer 3, excitability following Kcna1 overexpression at P7 has also been recorded (see answer to Reviewer 3 point 2).

With regard to in vivo recordings in young mice, this indeed is technically highly challenging, and in fact has to our knowledge never been done (reviewed in Blumberg *et al.*, 2015). Findings in rats may not be transposable to mice since the whisker-to-barrel pathway has significant species-specific properties (e.g. L4 cells invade barrel hollows in rats while they are confined to barrel walls in mice; e.g. Welker and Woolsey, 1974). **Ages at which experiments were performed are now highlighted**

in the text and figures and a sentence has been added on the developmental dynamics of the increased excitability.

2. For both dendrite morphology analyses and physiological analyses, it appears that the authors used the same controls many times. Did the authors use statistics for “multiple comparison”? Otherwise, it is not appropriate. If they used multiple comparison, the statistic method they used should be stated clearly.

Yes, corrections for multiple comparisons were performed in all analyses (one-way ANOVA with Tukey post-hoc test). These data are summarized in Fig. S2 (morphological analyses) and in the new Supplementary Table 2 (electrophysiological analyses, see Reviewer 3 Point 1).

3. Even if the authors used multiple comparison statistics properly, I still have a concern. Because in utero electroporation at E12.5 is technically difficult, it is not easy to label neurons consistently. If a neuron is brighter than another neuron, even if both neurons have the same dendritic length, dendritic length of the former could be estimated longer. Also because the morphologies of individual neuron dendrites are highly variable, it is very important to avoid bias in selecting neurons for analyses. A useful way to solve these problems would be to prepare distinct control for each analysis and conduct experiments in a blind fashion. I do not think that the authors can repeat all the experiments again with this way. But at least, some most critical experiments should be repeated for a new sample/control set in a blind fashion to confirm their conclusions.

The reviewer raises the possibility that there might be a biased selection of analyzed cells across conditions (i.e. that cells chosen for analysis might not be representative of labelled cells as a whole).

Only one criterion was used to select cells for quantification: no overlap with other labelled neurons; specifically, no overlap across primary dendrites.

Fluorescence intensity was not used as a criterion; indeed, immunohistochemistry against GFP was performed in all conditions, such that signal intensity was high across cells.

To directly examine whether signal intensity affects dendritic length measurements, we performed an additional analysis within and across conditions examining the relationship between soma brightness and total dendritic length (which

we considered would be the most likely morphological parameter to be affected by a decrease in signal).

As illustrated below, **there was no relationship between fluorescence and dendritic length, suggesting that the fluorescence signal was saturated in most cells and that cell brightness is not a limiting factor under our experimental conditions.**

Finally, confirming the lack of a potential bias, a blind re-analysis of our data following cell reshuffling confirmed our previous conclusions, as shown below. **Details of the analytical procedure are now provided in the Methods.**

4. It is generally thought that the NMDARs regulate circuit refinement (including dendritic maturation) via Hebbian-type mechanism. While, the authors claimed that NMDAR's role is just to suppress the membrane excitability and simple overexpression of Kir1.1 can rescue the NMDAR knockout phenotype. If it is the case, it would be very surprising. A problem of their analysis is that their conclusions rely on very simple parameters of dendritic morphology. They used only total dendrite length, number of primary dendrites, number of branch points, and Sholl analyses. Importance of dendritic morphology of VPM neurons is not just complexity. A more important aspect is whether dendrites confined to a single barreloid or not. The authors should analyze dendritic expansion in respect to barreloid patterns. Although ThGrin1KO and IONS mice do not have barreloids, in utero electroporation-based knockdown (knockout) and rescue could be possible.

In contrast to L4 neurons in cortical barrels, the dendritic tree of VPM neurons is not typically confined to a single barreloids (Chiaia *et al.*, 1991; Lavallée et Deschênes,

2004; Varga *et al.*, 2002), such that an effect on dendritic distribution would not be expected. **This point is now presented in the text.**

We have nonetheless now performed the suggested analysis, which shows no effect on distribution over a single vs. multiple barreloids (One-way ANOVA with Tukey post-hoc test; barreloids limits were defined based on VGlut2 IHC).

See also answer to Point 2 Reviewer 2, regarding the polarization of dendrites.

5. (Figure 1a, Supplementary Figure 1) Please show better images for GFP-labeled neurons. With these images, dendrite morphologies are not very clear. In addition, barreloid pattern should be shown along with GFP-labeled dendrites.

This is now done.

Minor comments:

1. In figures and/or figure legends, mouse age and sample numbers (both mouse number and cell number) used should be stated clearly. In particular, mouse age information for electrophysiology data is necessary in figures and figure legends.

This is now done.

2. (Figure 1) Please show an image of P2 GFP-labeled dendrites.

Done.

3. (Figure 1a right) What are white dotted lines?

The lines delineate barreloids. This is now mentioned in the figure legend.

4. (Figure 1a left) Brainstem-thalamus axons do not cross the midline. In this figure, whiskers should be positioned on the left side.

Brainstem-to-thalamus axons do cross the midline (see e.g. review in Erzurumlu and Gaspar, 2012), yet our schematic was somewhat misleading in its layout. This has now been updated for better clarity.

5. (Figure 1 legend) “(d) Dendritic maturation is DELAYED ----” How can the authors use the word, “DELAYED”? Without analyzing other ages such as at P2 and in adulthood, the authors cannot claim that the dendritic maturation is “DELAYED”. Similarly, in text, avoid using “delayed (e.g. Main text p.5)” and “precocious development (Main text p.6)”

Agreed. “Delayed” has been replaced with “Impaired” and “precocious” has been removed.

6. (Figure 2a, 2b,) What are red arrowheads?

The red arrowheads show the microdissected specimen and its original location. This is now mentioned in the figure legend.

7. (Figure 2) Mouse age and sample numbers should be clearly stated in the legend.

Done.

8. (Figure 3c) What are arrowheads? Describe the mouse number used for in situ (Fig.2, too).

The arrowheads Show the labeling in the VPM. This is now mentioned in the figure legend. Mouse numbers are now provided.

9. (Supplementary Figure 1) What are red dotted lines? Please specify which traces are corresponding to the GFP images.

The dotted lines delineate barreloids. This is now mentioned in the figure legend. Correspondence is now indicated.

10. (Supplementary Figure 3b) Expression outside the VB looks higher in TgGrin1KO than in WT. Why?

This reflects our choice of illustration (lowmag presented below), and was not systematically present across ages. A different layout is now shown.

11. (Methods) There are multiple lines of Sert Cre mice. Please specify mouse line that you used.

The Mouse line used is SertCre (B6.129(Cg)-Slc6a4tm1(cre)Xz/J, Jackson Laboratory, stock number 014554. This is now updated in the Methods.

12. (Method) What is pCAG-IRES-GFP? Please describe how the authors constructed or obtained this plasmid.

The plasmid was originally provided by Guillermina Lopez-Bendito. This is now mentioned in the text.

13. (Method) (Imaging and quantification) Please describe mouse number used for analyses.

This is now done.

Reviewer #2

In this manuscript, Frangeul et al. uses mouse somatosensory system to demonstrate that removing peripheral sensory input or deleting a subunit of NMDAR in thalamic projection neurons causes increased excitability and impaired maturation of dendritic morphology of these neurons and defective whisker-specific patterning of the thalamic nucleus. They further show that increased excitability is due to decreased expression of the potassium channel Kv1.1 and that *Kcna1* expression is epigenetically regulated by GADD45B, which is induced by NMDAR activation. These findings are novel and will be of great interest to the broad audience of developmental neurobiology.

To support the above conclusions, the authors used a wide variety of approaches including mouse genetics, in utero electroporation, analysis of gene expression, neuronal morphology and electrophysiology. Overall, the data are clearly presented and support their conclusions. In addition, experimental procedures are detailed enough and statistical analysis are appropriate. However, there are some missing experimental data or discussion, which makes it harder to think about this paper in the context of previously published work.

We thank the reviewer for his/her comments and now provide additional experimental data and discussion, as detailed below.

1. First, the authors show that barreloid formation in VPM is disrupted in *ThGrin1*KO mice as well as IONS mice, which has been published by others. However, whether the *Gadd45b* knockout or *Kcna1* knockdown mice have similar defects is not addressed. If the GADD45B-Kv1.1 pathway is an essential downstream pathway of NMDAR and is central to the role of NMDAR or peripheral input in the formation of whisker-specific patterning in the somatosensory system, it should be expected that *Gadd45b* knockout and *Kcna1* knockdown mice also show similar barreloid phenotypes as *ThGrin1*KO or INONS mice. Authors should show and discuss these data.

Knockdown of *Kcna1* via in utero electroporation cannot be used to study barreloid assembly because the fraction of cells affected per barreloid is too low to affect overall structure (which is also why electroporation of Cre^{GFP} in *Grin1*^{flox/flox} mice does not affect barreloid structure, in contrast to *ThGrin1*KO mice).

We have, however, been able to examine the brain of a *Kcna1*^{-/-} mouse (gift of Bruce Temple, U. Washington; Smart et al., 1998) and have also examined *Gadd45b*^{-/-} brains; neither mutant showed obvious impairments in whisker patterning (as illustrated below).

The persistence of barreloids and barrels in these two whole-body KOs may reflect at least two processes:

First, mosaic deletion approaches as performed here do not necessarily reflect the situation found in whole body knockouts (see e.g. Datwani et al, vs Mizuno et al for differences between global and mosaic manipulations of *Grin1* in L4 barrel neurons). Indeed, the former approach introduces a competitive advantage/disadvantage for manipulated cells, which is not the case in whole body knockouts, in which all cells are equally affected. Competition between manipulated and non-manipulated cells might be particularly relevant in studies of neuronal excitability, in which global neuronal silencing and local silencing have strikingly different effects (see e.g.

interocular interactions for visual map development, Zhang et al., 2011; or effects of unilateral vs. bilateral silencing for interhemispheric connectivity, Suarez et al., 2014).

Second, regulatory mechanisms might come into play early in development to compensate for the missing gene. Indeed, in contrast to GRIN1, which is an essential subunit of NMDARs and which cannot be compensated for, two additional isoforms of GADD45 exist (a and g), which have been reported to compensate for each other (e.g. *Gadd45g* increases in the liver of *Gadd45b*^{-/-} mice Tian et al. J Clin Invest, 2011). Similarly, Kv1.1 is not an essential subunit of Kv1 channels: these channels are tetramers, and 7 other subunits exist for which compensatory transcriptional regulations have been reported (London et al., Circ Res, 2001).

These data are now presented as Supplementary Fig. 5 and discussed in the Discussion.

2. Second, the Zhang et al. (2014) paper used a similar strategy to knockout *Grin1* in thalamic neurons and found that pruning and strengthening of immature synapses are blocked in neurons without NMDARs. In addition, Lee et al., (2005) found that in *Grin1* KD mice, VPM neurons had longer dendrites with no orientation preference. How do these results compare with the findings in the current manuscript in which dendritic arborization is reduced and excitability is increased when either of the molecular pathway components is disrupted? It will be informative and feasible to analyze the symmetry of dendrites shown in Supplemental Fig.1.

The Lee *et al.* study from the Erzurumlu lab does not study VPM dendrites but instead focuses on VPM axon development in cortex-specific *Grin1*KO mice. We assume that the reviewer is referring to the Datwani *et al.* (2002) study from the same group in which the dendrites of L4 neurons are longer and lose their orientation preference in cortex-specific *Grin1*KO mice. Might a similar process might be at play in VPM neurons?

Confirming previous data (Chiaia *et al.*, 1991; Lavallée et Deschênes, 2004; Varga *et al.*, 2002 and see answer to Reviewer 1 point 4), **we did not find any orientation preference in our dataset under control conditions or under any experimental condition (Data presented below)**. Indeed, in contrast to L4 barrel neurons, VPM barreloid neurons do not orient their dendrites towards barreloids hollows (this point is now explicitly mentioned in the results).

Although VPM neurons and L4 barrel neurons belong to the same functional pathway, NMDAR activation appears to have a different effect on dendritic growth in these two cell types: loss of *Grin1* increases dendritic length in L4 neurons (Datwani

et al) but decreases it in VPM neurons (current study). These findings suggest that NMDARs can act on a variety of cellular differentiation programs, and modulate features of neuronal development and circuit formation in a cell type-specific manner.

This point is now discussed in the Discussion.

Figure Point 2: Top: The distribution of VPM neuron primary dendrites is symmetrical across conditions. Bottom: reconstruction of the primary dendrite orientation for the cells presented above.

Reviewer #3

This study by Frangeul et al. is an elegant investigation of the mechanisms involved in the control of dendritic patterning in mouse somatosensory thalamus during development. The authors demonstrate the NMDA receptor dependence and

interaction with KV1.1 that underlies input-dependent maturation and excitability of VPM neurons. The strength of the paper is the identification of epigenetic regulation of this process. The electrophysiological characterization of the excitability changes and relation to dendritic structure is very important as well, but lacking in thoroughness on the biophysical level. My critique relates primarily to the latter shortcomings, as well as places where the methods presented are insufficient for other researchers to replicate these findings, and finally issues that should be clarified with text revision.

Major.

1. The in vitro electrophysiological properties are woefully under-analyzed. In addition to the spike number measurements that the authors present, a number of important properties should be described. These include (but are not necessarily restricted to) resting membrane potential, input resistance, series resistance, action potential threshold (rheobase), action potential height and width.

We now provide additional information the electrophysiological properties of the cells (Supplementary Table 2). In contrast to spike frequency at P15 (Figs. 2c and 4a), which is increased in all experimental conditions compared to controls, none of the additional parameters examined showed systematic differences. AP width tends to decrease compared to control cells (which is compatible with a higher-frequency firing ability) and AP height tends to increase (which might reflect hyperexcitability), but these changes are not systematic across conditions. Thus, spiking frequency in response to a depolarizing pulse (called here excitability) appears to be the most sensitive measure of the changes in bioelectrical membrane properties following loss of Grin1 or IONS.

These data are now presented as Supplementary Table 2 and in the results.

2. The authors found that decreased branching is associate with hyper-excitability. With some manipulations, increased branching was found (shRNA; Kcna1 electroporation). If the authors want to solidify the relationship between branching and excitability, experiments should be performed in the neurons with increased branching.

In the Kcna1 electroporation experiment (Fig. 5), it appears the authors did not test the electrophysiological properties of the neurons and the conclusions are based on dendritic branching alone. Conclusions about the excitability of the Kcna1 electroporated neurons seem unsupported.

We agree with the reviewer that this is an important point. **We have now electroporated Kcna1 in hyperexcitable neurons following IONS. This reduced neuronal excitability to control levels, supporting our interpretation of the dendritic phenotype.**

This data is now presented in Supplementary Fig. 6a, and reproduced below.

3. While the authors use a nerve lesion experiment to show a relationship between sensory input and dendritic maturation in thalamic neurons, there is nothing done to measure properties of the synaptic input in a direct way, i.e., via stimulation of axons in slice recordings. Therefore, there is no information about how the branching patterns and excitability relate to properties of altered synaptic input. It is possible that the important changes in burstiness found in vivo could be due to presynaptic hyper-excitability (i.e. via increased excitability in trigeminal ganglion). It would be ideal if the authors could perform experiments to measure the properties of synaptic input to thalamic neurons to test whether they have changed. But at minimum, this topic needs to be thoroughly discussed as to whether the changes in branching and excitability measured in this study can be attributed to thalamic neurons themselves, or to presynaptic inputs from trigeminal ganglion or top-down inputs.

The impact of mosaic deletion of *Grin1* in the VPM on synaptic transmission has already been examined in a previous study (Zhang *et al.*, 2013), showing that in the absence of NMDARs, the pruning of PrV axons onto VPM neurons is decreased (i.e. more PrV inputs are detected using VGLUT2 staining). Interestingly, most of these inputs appear to be non-functional, since upregulation of AMPARs was disrupted at these synapses.

Our findings are compatible with this scenario, in that increased excitability could reflect a homeostatic “denervation hypersensitivity”, as alluded to in the discussion in the context of IONS. **We now expand on this idea in the text following the reviewer’s suggestion.**

With regard to the plasticity of top-down inputs onto thalamic neurons following peripheral lesions, only very little is known. Recently, we and others have shown that L5B input, which normally target higher-order nuclei are rewired onto visual thalamic neurons following enucleation (Frangéul *et al.*, 2016; Grant *et al.*, 2016), but whether this occurs in the somatosensory system remains to be tested.

This point has also been added in the discussion.

4. The Methods contain insufficient information for others to replicate the experiments.

A. In the ION section description, the authors state that it was performed 'as previously described', yet in the cited study, the methods do not describe the procedure, they only reference a paper from 1990. This is an inappropriate self-

citation that contains no information and should be removed. The ION procedure should be described in the current paper, as differences are bound to exist compared to a paper from a different lab from 27 years ago.

We now detail this procedure.

B. In the Imaging and quantification part, a link or reference should be provided for the Image J Sholl Analysis pug-in.

Done.

C. In the next paragraph, good luck to any researcher trying to replicate the SVM used here. The sentence 'The SVM model was constructed using dedicated R packages' leaves the reader with essentially or useful information about the SVM model or how it was constructed. More information needs to be provided.

We now provide more details on this topic.

Minor

1. In the experiments where electroporation is used to express Cre-GFP in *Grin1*-lox/lox mice (e.g., first part of Results), it should be stated that thalamic neurons are not the only target of *Grin1* deletion. The authors should state any potential issues related to upstream/downstream effects from potential indirect effects on thalamic neurons due to global deletion of *Grin1*. Specifically, the sentence '... genetically ablated these receptors in VPM neurons...' needs to state that it was not exclusively in these neurons.

In the case of in utero electroporations, thalamic neurons are in fact the only targets of *Grin* deletion, since electroporation is performed in the dorsal part of the third ventricle. Although a fraction of cells (~20%) are found in the PO and other thalamic nuclei, they were not analyzed in this study. Indirect effects of these exogenous cells onto VPM neurons is highly unlikely given the lack of nucleus-to-nucleus connectivity in the thalamus.

In the case of the *ThGrin1*KO, Sert Cre expression is highly enriched in the VPM (see Pouchelon et al, 2014) but also, to a lesser extent, in the PO and LGN. This is now mentioned in the main text as follows:

*For this purpose, we generated transgenic mice which lacked *Grin1* in VPM neurons (and, to a lesser extent, in the PO and LG nuclei), by crossing *Sert*^{Cre} mice with *Grin1*^{lox/lox} mice (henceforth referred to as *ThGrin1*KO mice; Supplementary Figs. 3a,b).*

2. The word 'rescue' in the abstract implies that the authors performed experiments to alter dendritic maturation after it was altered by sensory deprivation. This is not true because the genetic manipulation was performed before sectioning the nerve. The word 'enable' (or alternative) may better capture the nature of the experiment.

Agreed, the wording has been updated throughout the text (using “enable” or “prevent”)

3. I suggest changing the term 'proof-of-principle' in the last paragraph of Introduction. To my ear this sounds like the study was done only on a few neurons to

show the results 'in principle', as one might do in a methods paper. Consider 'model system' or alternative.

We now use the term “model population”

4. Results, first paragraph: The sentence 'NMDARs have been involved in..' Involved should be changed to 'implicated in' or alternative.

This is now done.

5. The authors identify Kv1.1 as a key target of the changes in excitability, but there is no discussion of changes in expression of other ion channels that could contribute to this phenomenon. The issue of whether Kv1.1 fully explains the effects should be addressed in the Discussion.

Agreed. This relates to the answer provided to Reviewer 2, Point 1 on compensatory mechanisms. We now discuss misregulation of other K^+ channels and their potential contribution to the current phenotype in the Discussion.

6. The meaning of the penultimate sentence in the Discussion is unclear ('... regulated according to context.').

This sentence has been removed and the discussion has been expanded on other topics (see above).

Reviewers' comments:

Reviewer #1 (Remarks to the Author):

The manuscript was improved greatly in this revised version.
But I still have a few concerns as follows:

(Major concerns)

1. The authors used the same term "ThGrin1KO" for both knockout mice and electroporation-based knockout cells. It is extremely confusing. As the authors mentioned clearly (Lines 204-206), these two cases are totally different. I think that the term "ThGrinKO" should be used only for Sert-Cre-mediated knockout mice. For the electroporation-based knockout cells, they should use a different term. I think that "electroporation-based Grin1KO" or simply "Grin1KO" may be better. For example, in Figure 2d, the authors used "shKcna1" for electroporation-based knock-down cells. In the same manner, "Grin1KO" cells should be appropriate for electroporation-based knockout cells.
2. Related to the above point, in order to claim that "Genetic ablation of NMDAR led to repression of Kv1.1, resulting in impaired dendritic maturation (lines 45-47)", the authors should demonstrate dendritic morphology of VPM neurons in ThGrin1KO mice, because they showed decreased Kv1.1 expression in ThGrin1KO mice but not in electroporation-based Grin1KO cells. It is also OK to show Kv1.1 expression level in electroporation-mediated Grin1 knockout cells.
3. (Fig.3C) In situ hybridization is not so quantitative.
4. (Figure 5 title) The term "rescues" is not appropriate in this context.
5. Some colors in the figures are difficult to be distinguished. For example, P7 WT and P7 shKcna1 of Fig.2d are both blue and too similar. Fig.2e, Fig.5, Sup Fig.4 also had problems in color. Probably the authors try to distinguish them by dotted or solid lines. But it is also difficult to judge because these lines are too short.
6. In addition to cell numbers, mouse numbers also need to be stated in figure legend.
7. (Sup Fig.6a legend) Please state the post-hoc analysis method. Mouse number used should also be stated.

(Minor points)

1. (Fig.1a three pictures in the right side) Which part of the low magnification picture (left) was enlarged in the picture in the most right picture?
2. (Line 87) "a increase" should read "an increase".
3. (Line 137) "NMDAR --- regulate" should read "NMDARs --- regulate".

4. (Line 153) "IONS" should be "infraorbital nerve section (IONS)".
5. (Line 180) "Table 2) was" should read "Table 2), was".
6. (Line 219) "These suggests" should read "These suggest".
7. (Line 273) "pad was" should read "pad, was".
8. (Lines 366, 388, 415) "pot-hoc" should read "post-hoc".

Reviewer #2 (Remarks to the Author):

The authors have now responded to both of the major concerns addressed for the first manuscript.

On the first point, they have now analyzed knockout mice for Gadd45ab and Kcna1.

Although neither Gadd45b ko or Kcna ko showed phenotypes in barrelloid formation, they propose two possible explanations for that are reasonable.

The authors have properly responded to the second point.

Reviewer #3 (Remarks to the Author):

The authors have satisfied my concerns, and in my view the paper is suitable for publication.

signed:

David J. Margolis

Reviewer #1 (Remarks to the Author):

The manuscript was improved greatly in this revised version.
But I still have a few concerns as follows:

We thank the Reviewer for these comments and his/her suggestions, and provide answers below.

(Major concerns)

1. The authors used the same term “ThGrin1KO” for both knockout mice and electroporation-based knockout cells. It is extremely confusing. As the authors mentioned clearly (Lines 204-206), these two cases are totally different. I think that the term “ThGrin1KO” should be used only for Sert-Cre-mediated knockout mice. For the electroporation-based knockout cells, they should use a different term. I think that “electroporation-based Grin1KO” or simply “Grin1KO” may be better. For example, in Figure 2d, the authors used “shKcna1” for electroporation-based knock-down cells. In the same manner, “Grin1KO” cells should be appropriate for electroporation-based knockout cells.

We agree with the Reviewer that the current terminology is confusing. To distinguish cells in knockout mice and electroporation-based knockout cells, we now refer to the former as ThGrin1KO and to the latter as Grin1KO^{ThEpor} throughout text and figures.

2. Related to the above point, in order to claim that “Genetic ablation of NMDAR led to repression of Kv1.1, resulting in impaired dendritic maturation (lines 45-47)”, the authors should demonstrate dendritic morphology of VPM neurons in ThGrin1KO mice, because they showed decreased Kv1.1 expression in ThGrin1KO mice but not in electroporation-based Grin1KO cells. It is also OK to show Kv1.1 expression level in electroporation-mediated Grin1 knockout cells.

We agree with the Reviewer that in this context (Point 1) the sentence as phrased in the abstract may be misunderstood. To emphasize the use of multiple approaches and avoid any confusion, we have rephrased our claim from:

“Genetic ablation of N-methyl-D-aspartate (NMDA) receptors during postnatal development led to an epigenetic repression of Kv1.1-type potassium channels, resulting in increased excitability and impaired dendritic maturation, as did lesions to whisker input pathways.”

to

“Using a combination of genetic approaches, we find that ablation of N-methyl-D-aspartate (NMDA) receptors during postnatal development leads to epigenetic repression of Kv1.1-type potassium channels, increased excitability, and impaired dendritic maturation. Lesions to whisker input pathways had similar effects.”

Importantly, a decrease in Kv1.1 expression levels in electroporation-mediated Grin1 knockout cells (which the reviewer suggests as a potential experimental aim) is supported by the fact that overexpression of Kv1.1 in Grin1KO^{ThEpor} restores dendritic complexity to normal levels (as shown in Fig. 5a). A note has been added in the Discussion to mention this point:

“Although Kcna1 expression levels have not been examined directly in Grin1KO^{ThEpor} neurons, overexpression of Kv1.1 restores the dendritic complexity of these cells to normal levels (Fig. 5a), supporting a downregulation of this transcript, as occurs in ThGrin1KO cells (Fig. 2b).”

3. (Fig.3C) In situ hybridization is not so quantitative.

Although ISH is not particularly quantitative or sensitive, this classical approach is reliable and well suited to identify large increases or decreases in the level of expression of a specific transcript. This is the case in our study: we provide here results for n = 3 mice in independent experimental replicates, which clearly demonstrate that Kcna1 expression is strongly and systematically repressed in Gadd45b^{-/-} compared to

WT. The number of replicates per ISH is indicated in the Figure Legends.

4. (Figure 5 title) The term “rescues” is not appropriate in this context.

The title of Fig. 5 has been modified to “Overexpression of Kv1.1 enables normal dendritic development of *Grin1*KO^{ThEpor} and VPM_{IONS} neurons”.

5. Some colors in the figures are difficult to be distinguished. For example, P7 WT and P7 shKcna1 of Fig.2d are both blue and too similar. Fig.2e, Fig.5, Sup Fig.4 also had problems in color. Probably the authors try to distinguish them by dotted or solid lines. But it is also difficult to judge because these lines are too short.

Colors have been modified for better readability. Line thickness has been increased in all figures so that dotted vs. solid lines can be better distinguished.

6. In addition to cell numbers, mouse numbers also need to be stated in figure legend. Done.

7. (Sup Fig.6a legend) Please state the post-hoc analysis method. Mouse number used should also be stated. Done.

(Minor points)

1. (Fig.1a three pictures in the right side) Which part of the low magnification picture (left) was enlarged in the picture in the most right picture? The magnified section is now highlighted in the low-mag picture.

2. (Line 87) “a increase” should read “an increase”. Done.

3. (Line 137) “NMDAR --- regulate” should read “NMDARs --- regulate”. Done.

4. (Line 153) “IONS” should be “infraorbital nerve section (IONS)”. Done.

5. (Line 180) “Table 2) was” should read “Table 2), was”. Done.

6. (Line 219) “These suggests” should read “These suggest”. Done.

7. (Line 273) “pad was” should read “pad, was”. Done.

8. (Lines 366, 388, 415) “pot-hoc” should read “post-hoc”. Done.

Reviewer #2 (Remarks to the Author):

The authors have now responded to both of the major concerns addressed for the first manuscript. On the first point, they have now analyzed knockout mice for *Gadd45ab* and *Kcna1*. Although neither *Gadd45b* ko or *Kcna* ko showed phenotypes in barreloid formation, they propose two possible explanations for that are reasonable.

The authors have properly responded to the second point.

We thank the Reviewer for his/her feedback on our work and for contributing to improve our manuscript.

Reviewer #3 (Remarks to the Author):

The authors have satisfied my concerns, and in my view the paper is suitable for publication.

We thank the Reviewer for his/her feedback on our work and for contributing to improve our manuscript.

REVIEWERS' COMMENTS:

Reviewer #1 (Remarks to the Author):

All of my concerns have been solved in this revised manuscript, and therefore I support the publication of this paper. I would also like to express my respect for the authors' efforts to improve the paper.